# Loss of autophagy impairs physiological steatosis by accumulation of NCoR1

Shun-saku Takahashi[1],*, Yu-Shin Sou[2],*, Tetsuya Saito[3], Akiko Kuma[4,5], Takayuki Yabe[6], Yuki Sugiura[7], Hyeon-Cheol Lee[8] ◉, Makoto Suematsu[7], Takehiko Yokomizo[8], Masato Koike[2] ◉, Shuji Terai[1], Noboru Mizushima[4,5] ◉, Satoshi Waguri[6], Masaaki Komatsu[3] ◉

**Lipid droplets (LDs) are dynamic organelles that store neutral lipids during times of energy excess, such as after a meal. LDs serve as an energy reservoir during fasting and have a buffering capacity that prevents lipotoxicity. Autophagy and the autophagic machinery have been proposed to play a role in LD biogenesis, but the underlying molecular mechanism remains unclear. Here, we show that when nuclear receptor co-repressor 1 (NCoR1), which inhibits the transactivation of nuclear receptors, accumulates because of autophagy suppression, LDs decrease in size and number. Ablation of *ATG7*, a gene essential for autophagy, suppressed the expression of gene targets of liver X receptor α, a nuclear receptor responsible for fatty acid and triglyceride synthesis in an NCoR1-dependent manner. LD accumulation in response to fasting and after hepatectomy was hampered by the suppression of autophagy. These results suggest that autophagy controls physiological hepatosteatosis by fine-tuning NCoR1 protein levels.**

## Introduction

Lipid droplets (LDs) are neutral lipid storage organelles that provide fatty acids (FAs) for energy production during periods of nutrient deprivation. These organelles, which emerge from the ER, also have a lipid buffering capacity that helps prevent lipotoxicity [1, 2]. Enzymes involved in triacylglycerol (TG) synthesis, such as diacylglycerol O-acyltransferase (DGAT), deposit neutral lipids in between the leaflets of the ER bilayer where neutral lipids demix and coalesce to form a structure called an oil lens. Thereafter, seipin and other LD biogenesis factors facilitate the growth of nascent LDs from this lens. LDs bud from the ER and grow through either fusion or local lipid synthesis [1, 2].

Apart from the selective degradation of LDs (lipophagy) [1, 3], there is growing evidence that autophagy or some element of the autophagic machinery plays an important role in LD biogenesis. First, there have been several independent observations of a reduction in LD number in knockout mice lacking autophagic components specifically in their hepatocytes [4, 5, 6, 7, 8]. Second, the autophagic machinery participates in LD formation in hepatocytes and cardiomyocytes [7, 9], and deletion of autophagy-related genes such as *Atg5* and *Atg7* in the mouse liver decreases the level of triglycerides in the liver [9] and impairs ketogenesis [8, 10]. Third, the loss of Fip200, an autophagy initiation factor, in mouse livers causes inactivation of nuclear receptors, liver X receptor α (LXRα), and peroxisome proliferator-activated receptor α (PPARα). These receptors play important roles in FA synthesis and oxidation, respectively [6]. Therefore, their inactivation blocks liver steatosis under physiological fasting and high-fat diet conditions [6]. Fourth, the supply of lipids provided through autophagy is required to replenish triglycerides in LDs [11], which provide molecules for FA oxidation. Fifth, the biogenesis of LD from FA supplied by starvation-induced autophagy prevents the lipotoxic effects of acylcarnitine [12], which disrupts mitochondrial membrane potential and mitochondrial function. However, whether autophagy participates in LD biogenesis directly and which step(s) within the process of LD biogenesis is affected by autophagy both remain unclear.

In this Research Article, we show that autophagy regulates FA and TG synthesis at the transcriptional level by fine-tuning the levels of nuclear receptor co-repressor 1 (NCoR1), a negative regulator of nuclear receptors, including LXRα, and that defective autophagy impairs physiological steatosis both under fasting conditions and after hepatectomy.

[1]Division of Gastroenterology and Hepatology, Niigata University Graduate School of Medical and Dental Sciences, Chuo-ku, Niigata, Japan   [2]Department of Cell Biology and Neuroscience, Juntendo University Graduate School of Medicine, Bunkyo-ku, Tokyo, Japan   [3]Department of Physiology, Juntendo University Graduate School of Medicine, Bunkyo-ku, Tokyo, Japan   [4]Department of Biochemistry and Molecular Biology, Graduate School and Faculty of Medicine, the University of Tokyo, Bunkyo-ku, Tokyo, Japan   [5]Department of Physiology and Cell Biology, Tokyo Medical and Dental University, Tokyo, Japan   [6]Department of Anatomy and Histology, Fukushima Medical University School of Medicine, Hikarigaoka, Fukushima, Japan   [7]Department of Biochemistry, Keio University School of Medicine, Tokyo, Japan   [8]Department of Biochemistry, Juntendo University Graduate School of Medicine, Tokyo, Japan

Correspondence: mkomatsu@juntendo.ac.jp
*Shun-saku Takahashi and Yu-Shin Sou contributed equally to this work

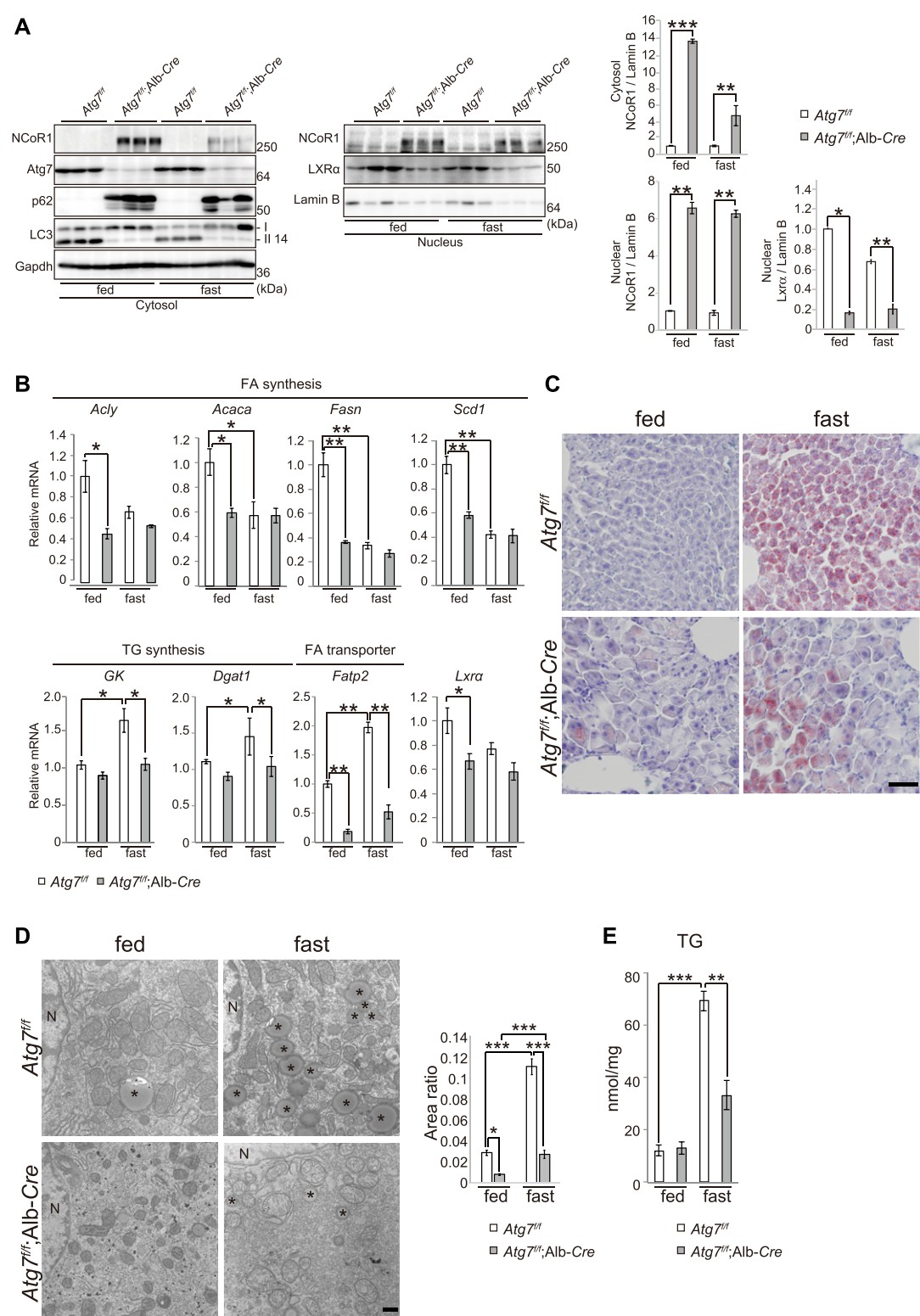

**Figure 1. Fasting-induced hepatosteatosis is suppressed by loss of autophagy.**
**(A)** LXRα and NCoR1 levels in *Atg7*-deficient mouse livers. Nuclear and cytoplasmic fractions, were prepared from the livers of 5-wk-old *Atg7*^f/f^ (n = 3) and *Atg7*^f/f^;Alb-*Cre* (n = 3) mice under both fed and fasting conditions. These were subjected to immunoblotting using the indicated antibodies. Signal intensities of cytoplasmic NCoR1 and Gapdh and nuclear NCoR1, LXRα, and lamin B were measured by densitometry and were subtracted that of background. Bar graphs indicate the average values of the indicated cytoplasmic and nuclear proteins relative to Gapdh and lamin B, respectively. The average value of the *Atg7*^f/f^ mice was set as 1. Data are means ± SE. *P < 0.05, **P < 0.01, and ***P < 0.001 as determined by Welch's *t* test. **(B)** Gene expression of proteins related to FA and TG syntheses in *Atg7*-deficient livers. Total RNA was prepared

# Results

## Impairment of fasting-induced hepatosteatosis in liver-specific *Atg7*-knockout mice

NCoR1 is an autophagy-specific substrate (10, 13) and serves as a scaffold that facilitates the interaction of several docking proteins to fine-tune the transactivation of nuclear receptors such as LXRα and PPARα (14, 15). The interaction of NCoR1 with nuclear receptors and histone deacetylases is vital for nuclear receptor–mediated down-regulation of gene expression. Interestingly, LXRα and PPARα, both of which are negatively regulated by NCoR1, play opposing roles in lipid metabolism. Specifically, LXRα serves anabolic roles (FA and TG syntheses), whereas PPARα serves a catabolic role (β-oxidation). To determine whether NCoR1 accumulation due to autophagy suppression has an impact on LD biogenesis, we used hepatocyte-specific *Atg7*-knockout mice, *Atg7$^{f/f}$*;Alb-*Cre* mice. The conversion of LC3-I to LC3-II was completely inhibited by the loss of *Atg7* (Fig 1A), and p62/SQSTM1 (hereafter referred to p62), another autophagy-specific substrate, accumulated in mutant livers (Fig 1A), implying that autophagy was impaired. In agreement with previous reports (10, 13), we verified that NCoR1 accumulates in both the nuclear and cytoplasmic fractions prepared from livers of *Atg7$^{f/f}$*; Alb-*Cre* mice (Fig 1A). Fasting decreased NCoR1 in both fractions from mutant livers, but levels of this protein were still higher than in control livers (Fig 1A). It has been reported that ubiquitination by a F-box-like/WD repeat–containing protein, TBLR1 directs NCoR1 into the proteasomal degradation and favors the exchange of corepressors for coactivators (16, 17). Thus, the ubiquitin-proteasome and autophagy-lysosomal pathways, both may contribute to degradation of NCoR1.

The expression of genes encoding enzymes involved in FA synthesis, including *ATP citrate lyase* (*Acly*), *acetyl-CoA carboxylase* (*Acaca*), *fatty acid synthase* (*Fasn*), and *stearoyl-CoA desaturase* (*Scd1*), which is regulated by LXRα, was markedly suppressed in the livers of *Atg7$^{f/f}$*;Alb-*Cre* mice under fed conditions (Fig 1B). Under fasting conditions, transcript levels of enzymes related to FA synthesis in control livers decreased to a similar extent as those in mutant livers (Fig 1B). Remarkably, although the genes encoding enzymes related to TG synthesis, such as *glycerol kinase* (*Gk*) and *diacylglycerol O-acyltransferase* (*Dgat1*), and a transporter of FAs, *fatty acid transport protein 2* (*Fatp2*), were up-regulated upon fasting, such induction was hardly observed in livers of *Atg7$^{f/f}$*;Alb-*Cre* mice (Fig 1B). The level of *Lxrα* mRNA was lower in mutant livers than in control livers (Fig 1B), consistent with the idea that LXRα regulates its own expression (18). We verified that the level of nuclear LXRα protein in mutant livers was significantly lower compared with that of control livers (Fig 1A). Because the autophagic turnover of NCoR1 is necessary for effective β-oxidation in response to fasting

(10, 13), these results suggest that under fasting conditions, both the catabolism (β-oxidation) and anabolism (TG synthesis) of FAs are primed by NCoR1 degradation. In fact, LDs detected by Oil Red O staining and electron microscopy showed that fasting-induced hepatosteatosis was suppressed by the loss of *Atg7* (Fig 1C and D). Consistent with the morphological analyses, the amount of TG in control livers increased upon fasting, but such increase was milder in mutant livers (Fig 1E). Similarly, fasting-dependent hepatosteatosis was blocked in livers of *Atg5$^{f/f}$*;Mx1-*Cre* mice one to 2 wk after intraperitoneal injection of polyinosinic-polycytidylic acid (pIpC), which induced liver-specific deletion of *Atg5*, another gene essential for autophagy (Fig S1A).

## Impairment of partial hepatectomy-induced hepatosteatosis in liver-specific *Atg7*-knockout mice

After partial (70%) hepatectomy, the remnant liver recovers to its original liver weight within approximately 1 wk after hypertrophy of hepatocytes and about two rounds of cell division (19). The liver shows a transient and prominent accumulation of FAs 1 d after resection (20, 21), which supports rapid cell division and tissue regrowth (22). Next, we investigated whether autophagy is also involved in LD biogenesis in hepatocytes after hepatectomy. To this end, we carried out a 70% hepatectomy on the livers of *Atg7$^{f/f}$* and *Atg7$^{f/f}$*;Alb-*Cre* mice and followed them until 168 h after hepatectomy. The blood level of free FAs in control *Atg7$^{f/f}$* mice gradually decreased after 70% hepatectomy, was at the lowest level at 18–24 h, and recovered 96–168 h after the hepatectomy (Fig 2A). In contrast, such fluctuation was not observed in mutant *Atg7$^{f/f}$*;Alb-*Cre* mice (Fig 2A), suggesting the impairment of free FA uptake from blood in mutant hepatocytes.

Whereas the expression of the FA transporter genes, *Fatp2* and *Fatp5*, in control livers was maintained up to 24 h after the hepatectomy, their expression levels in mutant livers were markedly decreased throughout the time course (Fig 2B). Moreover, we observed that in control livers, the transcription of genes that encode rate-limiting enzymes related to TG synthesis such as *Dgat1* and *Gk* was dramatically increased up to 24 h. This induction was suddenly terminated 48 h after hepatectomy (Fig 2B). In contrast, the expression of enzymes involved in FA synthesis dropped to its lowest level 24 h and only recovered to or exceeded the basal level 96–168 h after hepatectomy (Fig 2B). In mutant livers, expression of almost all genes involved in both FA and TG synthesis, except *Acly*, were suppressed, especially during the early recovery phase after hepatectomy (Fig 2B). These results suggest that steatosis is defective in autophagy-deficient livers.

Oil Red O staining indicated hyperaccumulation of LDs in control hepatocytes 24 h after hepatectomy and near recovery 48 h after

from the livers of 5-wk-old *Atg7$^{f/f}$* (n = 4) and *Atg7$^{f/f}$*;Alb-*Cre* (n = 4) mice under both fed and fasting conditions. Values were normalized against the amount of mRNA in the livers of *Atg7$^{f/f}$* mice under fed conditions. Real-time PCR analyses were performed as technical duplicates on each biological sample. Data are means ± SE. *P < 0.05 and **P < 0.01 as determined by Welch's t test. **(C)** Oil Red O staining. Cryosections were prepared from livers of 5-wk-old *Atg7$^{f/f}$* and *Atg7$^{f/f}$*;Alb-*Cre* mice under both fed and fasting conditions and subjected to Oil Red O staining. Bar: 50 μm. **(D)** Electron microscopy. **(C)** Representative electron micrographs of hepatocytes from the same genotype mouse as in (C) are shown. Ratio of LD area was measured and plotted in the right graph. Data are means ± SE. *P < 0.05 and ***P < 0.001 as determined by Welch's t test. Asterisks: LD, N: nucleus, Bar: 500 nm. **(B, E)** Liver triglyceride (TG) in mice described in (B). Data are means ± SE. **P < 0.01 and ***P < 0.001 as determined by Welch's t test.
Source data are available for this figure.

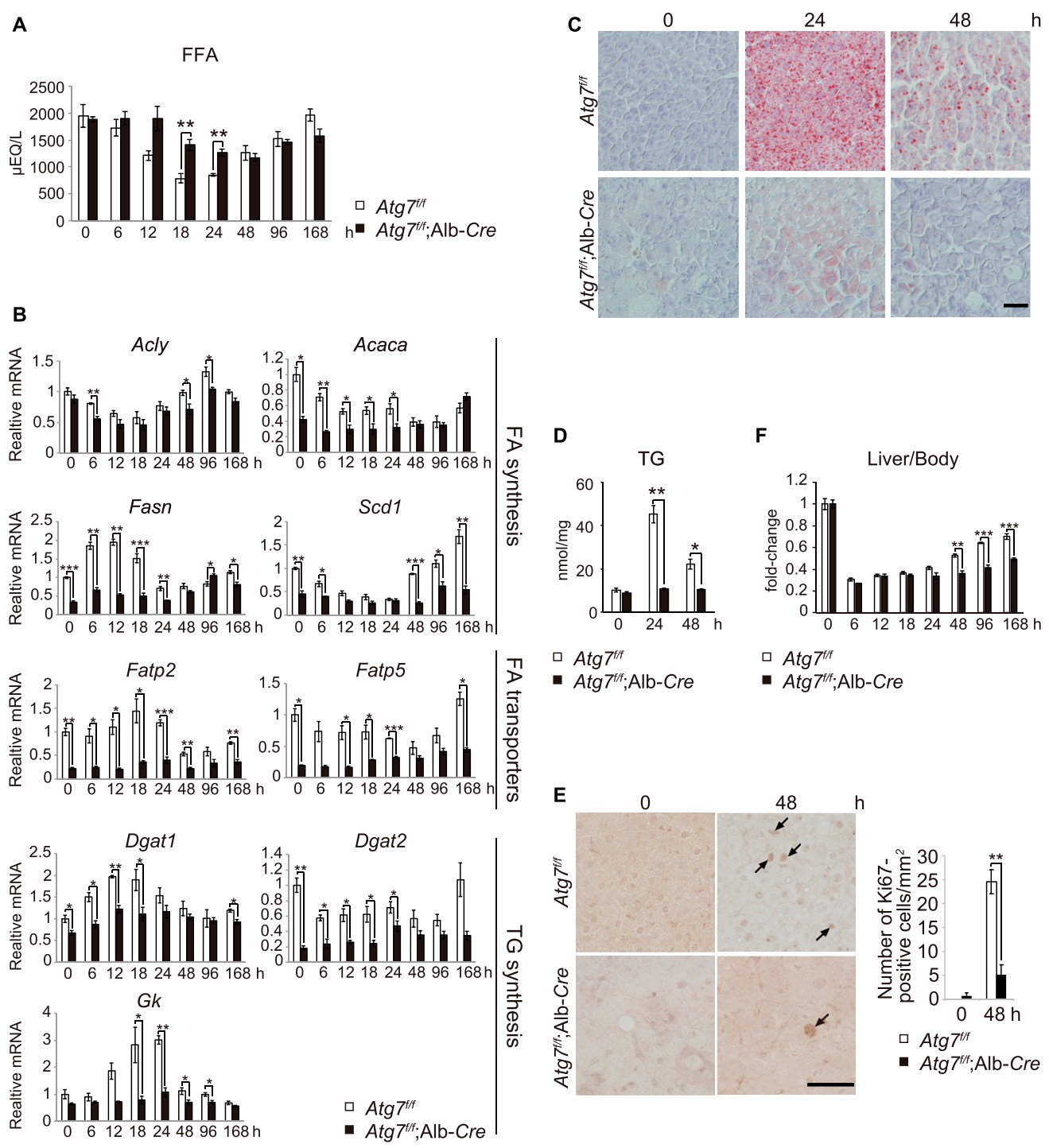

**Figure 2. Hepatectomy-induced hepatosteatosis is suppressed by loss of autophagy.**
**(A)** Blood-free FA. 5-wk-old $Atg7^{f/f}$ (n = 4) and $Atg7^{f/f}$;Alb-*Cre* (n = 3) mice were subjected to 70% hepatectomy. Subsequently, blood-free FA was measured at the indicated time points after hepatectomy. Data are means ± SE. **$P < 0.01$ as determined by Welch's $t$ test. **(B)** Gene expression of proteins related to FA and TG syntheses in $Atg7$-deficient livers after partial hepatectomy. **(A)** Total RNAs were prepared from mouse livers described in (A). Values were normalized against the amount of mRNA in $Atg7^{f/f}$ livers immediately after the hepatectomy. RT qPCR analyses were performed using cDNAs prepared from 5-wk-old $Atg7^{f/f}$ (n = 4) and $Atg7^{f/f}$;Alb-*Cre* (n = 3) mice in duplicate. Data are means ± SE. *$P < 0.05$, **$P < 0.01$ and ***$P < 0.001$ as determined by Welch's $t$ test. **(C)** Oil Red O staining. 5-wk-old $Atg7^{f/f}$ and $Atg7^{f/f}$;Alb-*Cre* mice were subjected to 70% hepatectomy. Liver sections were prepared at 0, 24, and 48 h after the hepatectomy and stained with Oil Red O. Data are representative of three separate experiments. Bar: 50 $\mu$m. **(A, D)** Liver triglyceride (TG) in mice described in (A). Data are means ± SE. *$P < 0.05$ and **$P < 0.01$ as determined by Welch's $t$ test. **(E)** Ki67-staining in liver paraffin sections prepared from 5-wk-old $Atg7^{f/f}$ and $Atg7^{f/f}$;Alb-*Cre* mice at 0 and 48 h after hepatectomy. Number of Ki-67–positive cells (arrows) were counted and plotted as the number per square millimeter in the right graph (n = 3 mice). Bars: 50 $\mu$m. Data are means ± SEM. *$P < 0.05$ and ***$P < 0.001$, as determined by Welch's $t$ test. **(A, F)** Liver weights (% per body weight) of mice described in (A). Data are means ± SE. **$p_{vov} < 0.01$ and ***$P < 0.001$ as determined by Welch's $t$ test.

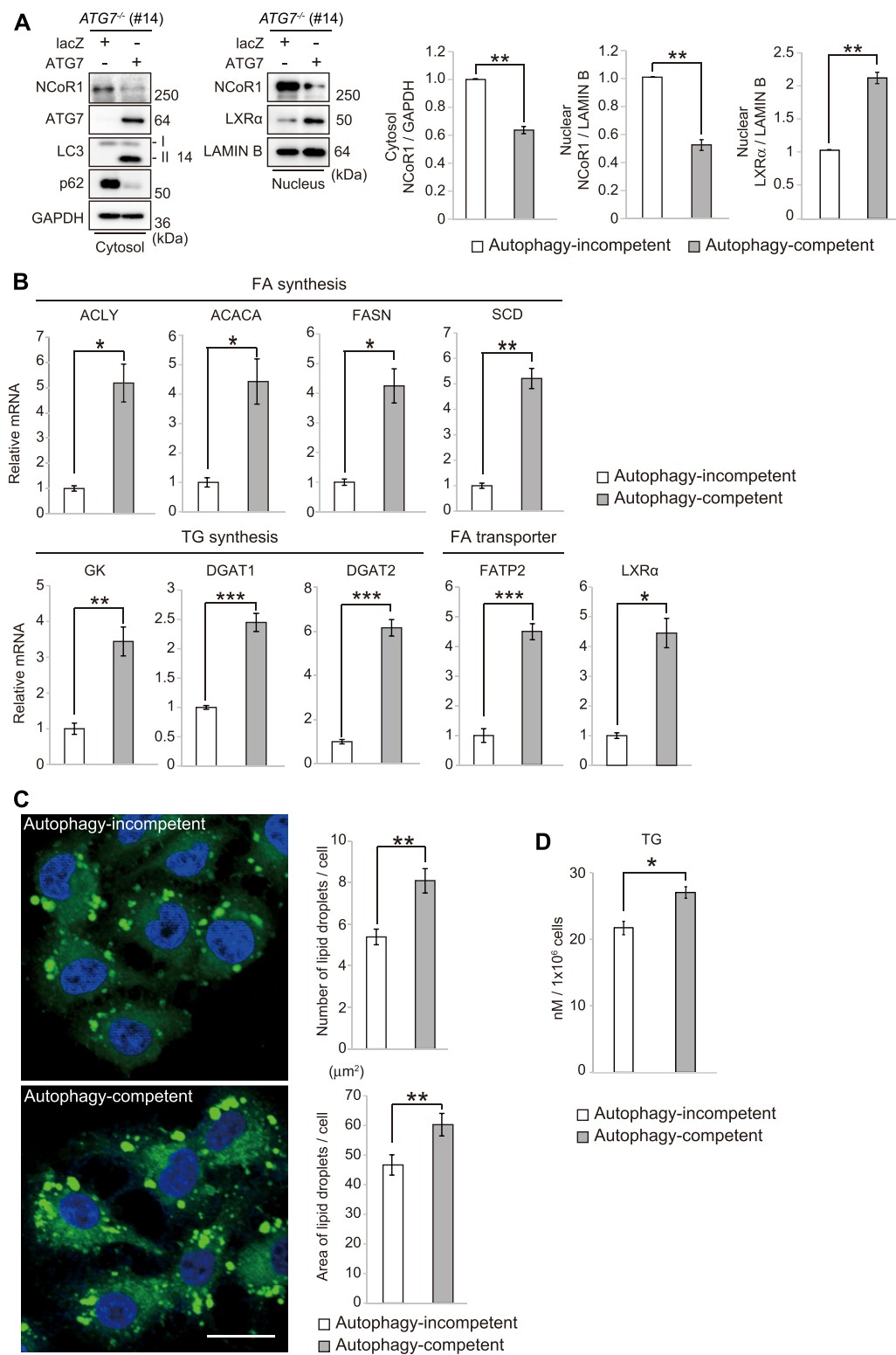

**Figure 3. Decreased levels of LDs in *ATG7*-deficient cells.**
**(A)** Immunoblot analysis. Either lacZ or ATG7 was expressed in *ATG7*-knockout HepG2 (#14) cells using an adenovirus system. 96 h after infection, both nuclear and cytoplasmic fractions were prepared and subjected to immunoblotting with the indicated antibodies. Data shown are representative of three separate experiments. Signal intensities of cytoplasmic NCoR1 and Gapdh and nuclear NCoR1, LXRα, and lamin B were measured by densitometry and were subtracted that of background. Bar graphs indicate the average values of the indicated cytoplasmic and nuclear proteins relative to Gapdh and lamin B, respectively. The average value of the *ATG7*-knockout HepG2 expressing lacZ was set as 1. Statistical analyses were performed using Welch's *t* test. Data are means ± SE. *$P < 0.05$ and **$P < 0.01$ as determined by

hepatectomy (Fig 2C). In contrast, such accumulation of LDs was not detectable in *Atg7*-deficient hepatocytes (Fig 2C). Consistent with those results, in control livers, the amount of TG markedly increased 24 h after hepatectomy and decreased 48 h after hepatectomy (Fig 2D). Such fluctuation was not observed in the case of mutant livers (Fig 2D). The hepatosteatosis that follows hepatectomy has been shown to play an essential role in hepatocyte proliferation (22). Therefore, we speculated that the loss of *Atg7* is accompanied by the impairment of liver regeneration after partial hepatectomy. Indeed, immunohistochemical analysis with anti-Ki67 antibody showed fewer Ki67-positive cells in mutant livers compared with control livers (Fig 2E). Liver weight per unit body weight in control mice indicated 80% recovery after 168 h compared with the weight before hepatectomy (Fig 2F). Recovery was observed even in mutant mice, but to a much lesser extent (Fig 2F).

### NCoR1 degradation through autophagy is necessary for increased level of LDs

Ultimately, we sought to elucidate the molecular mechanism by which loss of autophagy impairs LD accumulation and used HepG2 cells lacking *ATG7* (Fig S2). We expressed either *lacZ* or *ATG7* in *ATG7*-knockout HepG2 cells (#14) and compared the resulting phenotypic differences. As shown in Fig 3A, the conversion of LC3-I to LC3-II was restored in $ATG7^{-/-}$ HepG2 cells by the expression of *ATG7* but not *lacZ*, and the level of p62 protein decreased upon overexpression of *ATG7* but not *lacZ*. These results support the idea that autophagy is restored in $ATG7^{-/-}$ HepG2 cells expressing *ATG7* (autophagy-competent) but not in cells expressing *lacZ* (autophagy-incompetent). Indeed, both the nuclear and cytoplasmic NCoR1 levels were lower in autophagy-competent HepG2 cells than in incompetent cells (Fig 3A). In contrast, we observed a higher level of nuclear LXRα protein in autophagy-competent cells (Fig 3A). The expression of LXRα target genes in autophagy-competent cells was much higher than in incompetent cells (Fig 3B). BODIPY-staining revealed that LDs still form in autophagy-incompetent cells (Fig 3C), but the number and size of LDs were significantly smaller than those in autophagy-competent cells (Fig 3C). In agreement with these results, we found that the amount of TG in autophagy-competent cells was higher than in incompetent cells (Fig 3D).

Next, we investigated whether NCoR1 accumulation in autophagy-incompetent cells directly affects the level of LDs. The reduced level of nuclear LXRα in autophagy-incompetent cells was increased by the knockdown of *NCoR1* (Fig 4A). *NCoR1* depletion restores gene expression of most LXRα targets in autophagy-incompetent cells (Fig 4B). Unexpectedly, the transcription of some LXRα targets in autophagy-incompetent cells, including *Fatp2* and *Dgat1*, did not increase after *NCoR1* ablation (Fig 4B), probably because of partial compensation by NCoR2, an NCoR1 family protein (23). *NCoR1* knockdown in autophagy-

incompetent cells had little effect on the size and number of LDs (Fig 4C) because *NCoR1* ablation enhances both the anabolism and catabolism of FAs (14, 15). Regardless, we confirmed that the amount of TG in autophagy-incompetent cells was restored to a significant extent by silencing *NCoR1* (Fig 4D). On balance, these results suggest that NCoR1 accumulation due to defective autophagy suppresses LXRα transactivation, resulting in the impairment of FA and TG syntheses and of LD formation.

## Discussion

Our finding differs from a prior report that mouse hepatocytes lacking *Atg7* increase the size and the number of LDs (3). The main difference in experimental settings between this prior study and ours is the age of the genetically modified $Atg7^{f/f}$Alb-*Cre* mice. While we used 5-wk-old $Atg7^{f/f}$;Alb-*Cre* mice, Singh R et al (3), used 4-mo-old mice. Under conditions where the regeneration of mature hepatocytes is defective, such as the lack of β-catenin, hepatic oval cells proliferate and differentiate into hepatocytes and cholangiocytes, both of which replace the liver mass with aging (24). During active proliferation, most hepatic progenitor cells derived from oval cells undergo maturation arrest and become dedifferentiated, but these progenitor-derived immature hepatocytes possess a high potential for developing into liver tumors. Hepatocyte-specific ablation of β-*catenin*, in fact, promotes tumorigenesis (24). Likewise, the long-term suppression of autophagy in mouse livers is accompanied by tumorigenesis (25, 26), suggesting that autophagy-defective hepatocytes may lose the ability to regenerate and that the hepatic oval cells may compensate it. Remarkably, both hepatocytes and cholangiocytes differentiated from oval cells express albumin at negligible levels (24). In fact, we observed the presence of hepatocytes that lack p62-positive structures mosaically in aged $Atg7^{f/f}$; Alb-*Cre* mice (i.e., autophagy-competent hepatocytes) (data not shown). Thus, we speculate that the livers of aged $Atg7^{f/f}$;Alb-*Cre* mice are partially composed of Atg7-intact hepatocytes derived from oval cells and that such hepatocytes accumulate LDs aggressively to compensate for the dysfunction of *Atg7*-deficient hepatocytes. Indeed, some perivenous hepatocytes in 5-mo-old $Atg7^{f/f}$;Alb-*Cre* mice contained many LDs (Fig S1B).

In the present study, we showed prominent accumulation of NCoR1 protein and impairment of lipogenesis in *Atg7*-knockout mice livers. As already described, long-term suppression of autophagy in mouse livers causes adenomagenesis, but does not progress to malignancy. However, the exact molecular mechanism still remains unclear. Remarkably, decreased expression of NCoR1, focal deletion of 17p11.2 containing *NCoR1* and mutations of *NCoR1* have been specified in human hepatocellular carcinoma (27, 28, 29).

---

Welch's *t* test. **(B)** RT-qPCR analysis. **(A)** Total RNAs were prepared from cells described in (A). Values were normalized against the amount of mRNA in *ATG7*-knockout HepG2 cells (#14) expressing *lacZ* (Autophagy-incompetent). RT qPCR analyses were performed using cDNAs prepared from autophagy-incompetent and competent HepG2 cells (n = 3) in duplicate. Data are means ± SE. *$P < 0.05$ and **$P < 0.01$ as determined by Welch's *t* test. **(C)** BODIPY staining. Either lacZ or ATG7 was expressed in *ATG7*-knockout HepG2 cells (#14) using an adenovirus system. 96 h after infection, the cells were stained by BODIPY. Bars: 20 μm. The number and size of LDs were quantified by CellInsight CX5 High-Content Screening Platform. Statistical analyses were performed using Welch's *t* test. Data are means ± SE. **$P < 0.01$ as determined by Welch's *t* test. **(D)** Triglyceride levels. **(A)** Lysates were prepared from cells described in (A), and the concentration of TG in each lysate was determined by using the Abcam Triglyceride Assay Kit. Statistical analysis was performed using Welch's *t* test. Data are means ± SE. *$P < 0.05$ as determined by Welch's *t* test.

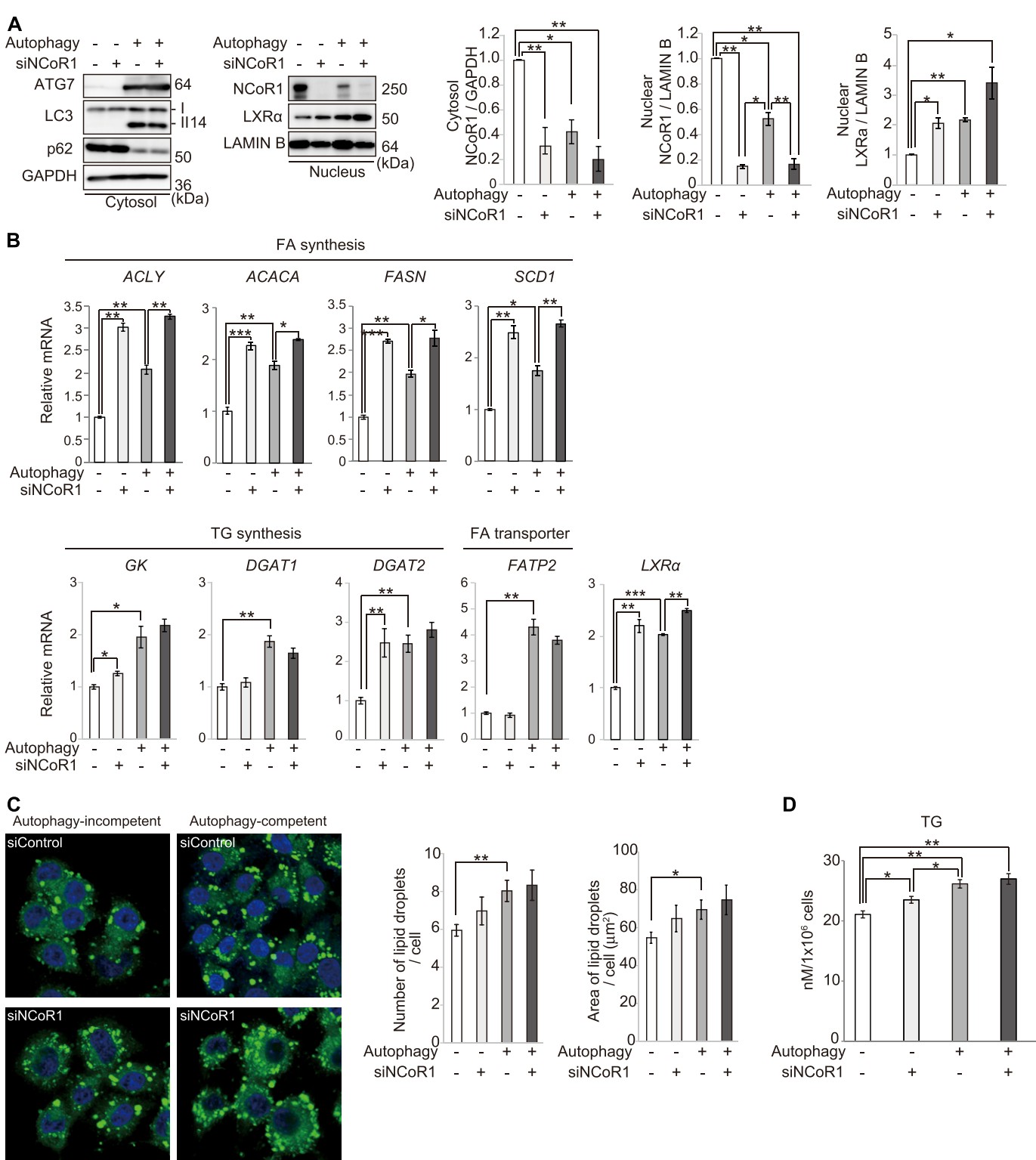

**Figure 4. Knockdown of *NCoR1* restores decreased levels of LDs in *ATG7*-deficient cells.**
**(A)** Immunoblot analysis. *ATG7*-knockout HepG2 (#14) cells were infected with adenovirus for lacZ (Autophagy−) or ATG7 (Autophagy+) expression. The cells were simultaneously treated with either *NCoR1*-specific or scrambled control siRNA. 48 h after infection, the cells were re-treated with either *NCoR1*-specific or control siRNA and cultured for an additional 48 h. Thereafter, both nuclear and cytoplasmic fractions were prepared and subjected to immunoblotting with the indicated antibodies. Data are representative of three separate experiments. Signal intensities of cytoplasmic NCoR1 and GAPDH and nuclear NCoR1, LXRα, and lamin B were measured by densitometry and were subtracted that of background. Bar graphs indicate the average values of the indicated cytoplasmic and nuclear proteins relative to GAPDH and lamin B, respectively. The average value of the *ATG7*-knockout HepG2 expressing lacZ was set as 1. Statistical analyses were performed using Welch's *t* test. Data are means

Meanwhile, cancer cells activate de novo FA synthesis to provide essential structural components and substrates for the generation of signaling molecules, and lipid synthesis contributes to cellular processes linked to tumor progression (30). Therefore, resistance of the liver-specific *Atg7*-knockout mice to progress from benign adenoma to liver cancer (25, 26) might be due to persistent expression of NCoR1.

Autophagy provides a substantial amount of FAs through the degradation of organelles under nutrient-deprived conditions (11). A robust influx of FAs from the blood into peripheral cells such as hepatocytes occurs under fasting conditions or after hepatectomy (20, 21). The increased level of intracellular FAs due to starvation-induced autophagy and/or fasting-triggered influx provides fuel for β-oxidation to produce energy; however, because β-oxidation intermediates such as acylcarnitine have a cytotoxic effect (12), cells have to maintain the levels of intracellular FAs below certain limits. In fact, DGAT1-mediated TG synthesis under nutrient-deprived conditions, when the amount of intracellular FAs is excessive, is necessary to mitigate the lipotoxic cellular damage caused by acylcarnitine (12). It is worth noting that the loss of autophagy in mouse livers is accompanied by NCoR1 accumulation, resulting in increased levels of acylcarnitine, in particular under fasting conditions (10, 13). We conclude that fine-tuning of NCoR1 protein levels through autophagy regulates the level of LDs to mitigate lipotoxicity.

# Materials and Methods

### Cell culture

HepG2 cells were grown in DMEM containing 10% FBS, 5 U/ml penicillin, and 50 µg/ml streptomycin. For knockdown experiments, HepG2 cells were transfected with 25 nM SMARTpool siRNA for *NCoR1* using DharmaFECT 1 (Thermo Fisher Scientific). *ATG7*-knockout HepG2 cells (10) were used in this study.

### Mice

*Atg7*$^{f/f}$ (31), *Atg7*$^{f/f}$;Alb-*Cre* (32), *Atg5*$^{f/f}$ (33), and *Atg5*$^{f/f}$;Mx1-*Cre* (33) mice in the C57BL/6 genetic background were used in this study. Mice were housed in specific pathogen–free facilities, and the Ethics Review Committees for Animal Experimentation of Niigata University, the University of Tokyo, and Jaunted University approved the experimental protocol. The concentration of liver triglycerides was determined using the Triglyceride Assay Kit, ab65336 (Abcam). Free FAs in plasma were analyzed by SRL (Tokyo, Japan). Fasting to 6-wk-old male mice was started at 8 PM and then continued during 24 h. After cervical dislocation of the fasted mice and control fed mice, their livers were removed. Partial hepatectomy (PHx) was performed in 6-wk-old male mice. Mice were anesthetized with an intraperitoneal injection of 0.05 ml/10 g body weight of a mixed anesthetic agents, consisting of medetomidine (0.06 mg/ml), midazolam (0.8 mg/ml), and butorphanol (1 mg/ml) in sterile normal saline and subjected to approximately 70% PHx by removing the left lateral and median lobes after midventral laparotomy. The mortality rate after 70% PHx was <1%. At indicated time points after 70% PHx, the mice were euthanized by cervical dislocation, and their livers were removed.

### Immunoblot analysis

Livers were homogenized in 0.25 M sucrose, 10 mM 2-[4-(2-hydroxyethyl)-1-piperazinyl]ethanesulfonic acid (Hepes) (pH 7.4), and 1 mM DTT. Nuclear and cytoplasmic fractions from livers and cultured cells were prepared using the NE-PER Nuclear and Cytoplasmic Extraction Reagents (Thermo Fisher Scientific). Samples were subjected to SDS–PAGE, and transferred to a polyvinylidene difluoride membrane thereafter (IPVH00010; Merck). Antibodies against LXRα (ab28478; Abcam; 1:500), PPARα (ab8934; Abcam; 1:500), NCoR1 (#5948S; Cell Signaling Technology; 1:500), Atg7 (013-22831; Wako Pure Chemical Industries; 1:1,000), p62 (GP62-C; Progen Biotechnik GmbH; 1:1,000), LC3B (#2775; Cell Signaling Technology; 1:500), Gapdh (MAB374; Merck Millipore Headquarters; 1: 1,000), and lamin B (M-20; Santa Cruz Biotechnology; 1:200) were purchased from the indicated suppliers. Blots were incubated with horseradish peroxidase-conjugated goat antimouse IgG (H+L) (115-035-166; Jackson ImmunoResearch Laboratories, Inc.), goat antirabbit IgG (H+L) (111-035-144; Jackson ImmunoResearch Laboratories, Inc.), or goat anti-guinea pig IgG (H+L) antibody (106-035-003; Jackson ImmunoResearch Laboratories, Inc.), and visualized by chemiluminescence. Band density was measured using the software MultiGauge V3.2 (FUJIFILM Corporation).

### RT-qPCR (real-time quantitative reverse transcriptase PCR)

Using the Transcriptor First-Strand cDNA Synthesis Kit (Roche Applied Science), cDNA was synthesized from 1 µg of total RNA. RT qPCR was performed using the LightCycler 480 Probes Master mix (Roche Applied Science) on a LightCycler 480 (Roche Applied Science). Signals from human and mouse samples were normalized against *GAPDH* and *Gusb* (ß-glucuronidase) mRNA, respectively. The sequences of primers used for gene expression analysis in either mouse livers or human cell lines are provided in Table S1.

### Histological examinations

Excised liver tissues were fixed by immersing in 0.1 M PB (pH 7.4) containing 4% paraformaldehyde and 4% sucrose. They were

---

± SE. *P < 0.05, and **P < 0.01 as determined by Welch's t test. **(B)** RT qPCR analysis. **(A)** Total RNAs were prepared from cells described in (A). Values were normalized against the amount of mRNA in *lacZ*-expressing *ATG7*-knockout HepG2 cells treated with control siRNA. RT qPCR analyses were performed using cDNAs prepared from each biological sample (n = 3) in duplicate. Data are means ± SE. *P < 0.05, **P < 0.01, and ***P < 0.001 as determined by Welch's t test. **(C)** BODIPY staining. **(A)** *ATG7*-knockout HepG2 (#14) cells were treated as described in (A). Bars: 20 µm. The number and size of LDs were quantified by CellInsight CX5 High-Content Screening Platform. Statistical analyses were performed using Welch's t test. Data are means ± SE. **(D)** Triglyceride levels. **(A)** Lysates were prepared from cells described in (A), and the concentration of TG in each lysate was determined using the Abcam Triglyceride Assay Kit. Statistical analysis was performed using Welch's t test. Data are means ± SE. *P < 0.05 and **P < 0.01 as determined by Welch's t test.

embedded in frozen optimal cutting temperature-compound or paraffin. The cryosections were stained with Oil Red O, and the paraffin sections were stained with rabbit anti-Ki67 antibody (clone SP6; Thermo Fisher Scientific) followed by N-Histofine simple stain mouse MAX PO kit (NICHIREI BIOSCIENCES) using 3,3′-diaminobenzidine. They were observed with a light microscope (BX51; Olympus). For quantification, Ki67-positive cells were counted in five rectangular regions (433 × 326 µm) per liver section of each mouse. Three to four mice were included in this analysis.

### Electron microscopy

Livers were fixed by immersing in 0.1 M PB containing 2% para-formaldehyde and 2% glutaraldehyde. They were post-fixed with 1% $OsO_4$, embedded in Epon812, and sectioned for observation with an electron microscopy (JM-1200EX; JEOL). For quantification, area ratio of LDs was measured in 20 hepatocytes for each mouse. Three mice were included in this analysis.

### Microscopy for cultured cells

For staining of LDs, the cells were incubated with 1 µg/ml BODIPY 493/503 (D3922; Thermo Fisher Scientific) diluted in PBS for 15 min and extensively washed with PBS. Finally, the cells were incubated for 5 min with 10 µg/ml of Hoechst 33342 diluted in PBS, washed with PBS and mounted on slides with Prolong Gold antifade mounting solution (Thermo Fisher Scientific). The cells were imaged using a confocal laser-scanning microscope (Olympus, FV1000) with a UPlanSApo ×60 NA 1.40 oil objective lens. Ten fields of cells were imaged for each experimental condition with a CellInsight CX5 High-Content Screening Platform (Thermo Fisher Scientific) using HCS Studio software.

### Statistical analysis

Values, including those displayed in the graphs, are means ± SE. Statistical analysis was performed using the unpaired $t$ test (Welch test). A $P$ value less than 0.05 was considered to indicate statistical significance.

## Supplementary Information

## Acknowledgements

We thank K Kanno (Fukushima Medical University) for his help in histological analyses. M Komatsu is supported by the Grants-in-Aid for Scientific Research on Innovative Areas (19H05706 to M Komatsu), the Japan Society for the Promotion of Science (an A3 foresight program, to M Komatsu and 18H02611 to M Komatsu), and the Takeda Science Foundation (to M Komatsu).

### Author Contributions

S-s Takahashi: data curation, formal analysis, and investigation.

Y-S Sou: data curation, formal analysis, validation, and investigation.
T Saito: data curation, formal analysis, and investigation.
A Kuma: data curation, formal analysis, and investigation.
T Yabe: formal analysis and investigation.
Y Sugiura: formal analysis and investigation.
H-C Lee: formal analysis and investigation.
M Suematsu: supervision.
T Yokomizo: supervision.
M Koike: supervision.
S Terai: supervision.
N Mizushima: conceptualization and supervision.
S Waguri: data curation, formal analysis, supervision, and investigation.
M Komatsu: conceptualization, data curation, formal analysis, supervision, funding acquisition, and writing—original draft, review, and editing.

### Conflict of Interest Statement

The authors declare that they have no conflict of interest.

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
