## [Reviewer comments · Life Science Alliance]

Life Science Alliance

Loss of autophagy impairs physiological steatosis by accumulation of NCoR1

Shun-saku Takahashi, Yu-Shin Sou, Tetsuya Saito, Akiko Kuma, Takayuki Yabe, Yuki Sugiura, Hyeon-Cheol Lee, Makoto Suematsu, Takehiko Yokomizo, Masato Koike, Shuji Terai, Noboru Mizushima, Satoshi Waguri, and Masaaki Komatsu

DOI: <https://doi.org/10.26508/lsa.201900513>

Corresponding author(s): Masaaki Komatsu, Juntendo University School of Medicine

Review Timeline:

Submission Date:	2019-08-02
Editorial Decision:	2019-08-22
Revision Received:	2019-11-24
Editorial Decision:	2019-12-17
Revision Received:	2019-12-18
Accepted:	2019-12-18

Scientific Editor: Andrea Leibfried

Transaction Report:

August 22, 2019

Re: Life Science Alliance manuscript #LSA-2019-00513-T

Dr. Masaaki Komatsu
Juntendo University School of Medicine
Department of Physiology
Hongo 2-1-1
Bunkyo-ku, Tokyo 113-8421
Japan

Dear Dr. Komatsu,

Thank you for submitting your manuscript entitled "Autophagy controls lipid droplet formation by fine-tuning NCoR1 levels" to Life Science Alliance. The manuscript was assessed by expert reviewers, whose comments are appended to this letter.

As you will see, the reviewers provide constructive input on how to strengthen your study and we would thus like to invite you to submit a revised version to us, addressing the individual concerns raised.

Importantly, lipid droplet formation depending on NCoR1 is not supported by the data (rev#2) and this issue needs to get either addressed experimentally (which may be challenging) or by adapting the manuscript text. The other suggestions made by all three reviewers seem straightforward to address via additional experiments/clarifications and text changes, but please do get in touch with us in case you would like to discuss the revision further.

Thank you for this interesting contribution to Life Science Alliance. We are looking forward to receiving your revised manuscript.

Sincerely,

B. MANUSCRIPT ORGANIZATION AND FORMATTING:

Reviewer #1 (Comments to the Authors (Required)):

Takahashi S. et al. have conducted a compelling study that demonstrated the requirement of

functional autophagy for lipid droplet biogenesis. In this work, authors have built on their recently reported observations on the essential role of selective autophagy for lysosomal degradation of NCoR1 transcriptional repressor. They have extended this work to show here that a build-up of NCoR1 in autophagy deficient cells results in inhibition of LXRA transcription factor with most striking consequences for lipid homeostasis. They have employed complementary approaches in autophagy deficient models in vivo (hepatic mouse mutants) and in cells to elegantly demonstrate that autophagic degradation of NCoR1 is essential for expression of LXRA targets in fatty acid and TG synthesis as well as transport. The highlight of this study is a demonstration that transient liver steatosis relies on autophagy in two physiological metabolic and growth paradigms such as fasting and acute proliferation in response to partial hepatectomy. Therefore, this study irrefutably demonstrated an essential role of functional autophagy not only for lipid degradation but also for lipid synthesis and lipid droplet biogenesis. In sum, it is an excellent work that exposes another functional facet of autophagy in physiology. Most importantly, it also opens to future mechanistic questions on how cells sense the needs in selective autophagy of transcriptional repressors to fine tune activity and expression of transcription factors and the relevance of these mechanisms to pathophysiology.

The manuscript written with great style that makes it agreeable to follow the story. Particularly, the introduction brings up known aspects of how autophagy affects lipid metabolism and clearly frames the main question of this work. I have just few minor suggestions:

- 1) Add an explanation to the immunoblot quantifications (e.g. Fig.1A) about how exactly comparisons are made (since relative values are presented should one condition be set as 1?). Also, it is surprising that quantification of Fig.1A for cytosolic NCoR1 shows so little difference unlike immunoblot capture that is presented.
- 2) If text length allows, it would be interesting to discuss that although nuclear pool of NCoR1 is protected from degradation in livers of autophagy deficient Atg7LKO mice, the fasting potentially induced degradation of cytosolic NCoR1 protein (judged from immunoblot presented on Fig.1A). Did total levels of NCoR1 change in livers of fasted mice?
- 3) Describe in the text the differences in LXRA expression shown on Fig1A.
- 4) Harmonize the labelling of LXRA throughout the manuscript.
- 5) Revise figure legends e.g. Fig.1A (remove remark about total homogenates), Fig.S1 (change "hepatocytes" to "liver sections"). Specify what it means that experiments were performed 3 times (technical or biological replicates). Indicate in Material and method section timing conditions of fasting, sacrifice and surgical procedure in mice.
- 6) If text length allows, it would be important to bridge in discussion the findings on the role of autophagy in lipid synthesis and hepatocyte proliferation. To this end, an induction of FA synthesis is the most marked response to siNCoR1 in autophagy deficient HepG2 cancer cell line (Fig. 4B). From the other hand, hepatocyte proliferation and expression of enzymes of FA synthesis were decreased in Atg7LKO mice after hepatectomy (Fig.2). Notably, an increased de novo lipid synthesis is also a metabolic hallmark of proliferating cancer cells while decreased expression of NCoR1 was reported in liver cancer (PMID: 21075309). In lights of these findings, it is plausible that resistance of the Atg7LKO mice to progress from benign adenoma to liver cancer (PMID: 21498569) might be due to persistent expression of NCoR1 repressor. Therefore, these novel findings presented by Takahashi S. et al. might also be pertinent for liver pathophysiology.

Reviewer #2 (Comments to the Authors (Required)):

This Short Report by Takahashi et al reports on defective lipid droplet biogenesis and reduced

hepatosteatosis in hepatocyte-specific Atg7 knockout mice. The authors have recently demonstrated that lipid oxidation was suppressed in the same mouse model (Saito et al., Nat Commun 2019). In this study, the suppression of lipid oxidation resulted from accumulation of nuclear receptor co-repressor 1 NCoR1, which suppressed PPARalpha activity and thereby decreased the expression of enzymes involved in beta-oxidation. In the current manuscript, the accumulation of NCoR1 is shown to result in the suppression of LXRalpha activity, decreasing the expression of enzymes involved in fatty acid and triglyceride synthesis. Overall, the current observations are in line with earlier findings and the known functions of LXRalpha in controlling lipogenic gene expression. NCoR1 is known to be a substrate for autophagy (Saito et al., 2019) and fine-tunes the transactivation of both PPARalpha and LXRalpha. Moreover, loss of autophagy initiation causes inactivation of both PPARalpha and LXRalpha (Ma et al., 2013). The present paper does not go beyond this mechanistically. The experiments are in general well executed and most of the results are sound. However, some of the major claims made are not well substantiated by the data. Please see specific points below.

1. The NCoR1 silencing experiments (Fig. 4) do not actually support a major role for NCoR1 in lipid droplet formation. First, no differences in LDs are observed (Fig. 4C) and the effect on TAG levels is marginal, with some 10% increase. The authors suggest that this is because NCoR1 ablation enhances both the anabolism and catabolism of FAs. If this is the case, the authors might obtain clearer effects e.g. by manipulating FA catabolism in NCoR1 silenced cells. Second, the authors have strictly speaking not studied the formation or biogenesis of LDs. Rather, the steady-state numbers and sizes of LDs were analyzed. As LD maintenance is also dependent on LD breakdown, one cannot conclude that it is the formation of lipid droplets that is affected. Thus, the data do not convincingly show that NCoR1 is important for LD formation and do not support these statements in the title and abstract.

2. The current findings are essentially opposite to those of Singh et al. (Nature 2009) that reports on increased size and number of lipid droplets in hepatocytes lacking Atg7. The authors speculate that the difference might be due to the different ages of animals: 5 weeks in the present study as compared to four months in Singh et al. This should be possible to reconcile easily, by aging the mice till 4 months and studying if the Atg7 deficient livers now become steatotic.

3. The differences in lipid droplet numbers and sizes quantified from cells do not match well with the images. For instance, the autophagy competent cells shown have many more lipid droplets than ten (or less), as indicated in the bar graphs. The images look as if lipid droplets may have fused (a common problem when using glycerol containing mounting media).

4. Figure 1 A repeats findings shown in Saito et al., 2019 and this should be clearly spelled out.

Reviewer #3 (Comments to the Authors (Required)):

Autophagy plays a role in both lipid droplet biogenesis and in lipid droplet turnover via lipophagy. The mechanisms underlying the role of autophagy in LD biogenesis are not fully understood. Recent work from Saito et al and Iershov et al (both 2019) have shown that autophagy can regulate lipid metabolism by promoting the turnover of NCoR1, a negative regulator of PPARa that induces expression of genes involved in fatty acid oxidation. However, NCoR1 also regulates LXRa which promotes lipid synthesis, and thus acts counter to PPARa. To address this dichotomy, the authors examined NCoR1 levels and activity in the liver of Alb-Cre:Atg7^{fl/fl} mice in which autophagy

is impaired. Consistent with Saito and Iershov, NCoR1 levels accumulated in the liver when ATG7 was knocked out and autophagy effectively reduced.

Acly, Fasn, Scd1 and Acaca are all LXRA target genes and were down-regulated in Atg7 deleted livers. Fasting reduced their expression in WT but not in the KO. Other targets GK, Dgat1, Fatbp2 that were upregulated in WT liver by fasting were not induced in the Atg7 KO livers. Lxra mRNA expression was also down consistent with Lxra being autoregulated. Basically, similar to effects on PPARa, autophagy inhibition caused NCoR1 to accumulate and to inhibit LXRA activity.

Interestingly, lipid droplet numbers in fasted livers were reduced by Atg7 deletion, based on ORO staining. Is this due to reduced formation of LDs? Suppression of lipogenesis and Lxra target genes was also detected in response to partial hepatectomy resulting in reduced liver size following regeneration. Knocking out ATG7 in HepG2 cells also caused NCoR1 to accumulate and LXRA to be inhibited, as determined by levels of LXRA target genes. This was rescued by knockdown of NCoR1 although this did not rescue the number and size of lipid droplets since NCoR1 influences both FA oxidation (via PPARa) and lipogenesis (shown here via LXRA).

Overall, this work is highly complementary to the studies of Saito et al and Iershov et al and extends them to show that autophagic regulation of NCoR1 also influences lipogenesis via LXRA. It does leave the conundrum of how and when NCoR1 may differentially regulate PPARa and LXRA in a manner modulated by autophagy. An aspect of the work that could be enhanced, is if the authors could measure rates of fatty acid oxidation and de novo lipogenesis in the control and Atg7 deleted hepatocytes to confirm that autophagy is indeed impacting both these processes via NCoR1 turnover. Work that is perhaps beyond the scope of this study is to examine mutants of LXRA that cannot bind NCoR1 - are these mutants resistant to the effects of autophagy inhibition. As it stands, the scope of the paper is limited but what is included is useful and well controlled data.

Referee's comments (italized)**Referee #1**General comments:

5 Takahashi S. et al. have conducted a compelling study that demonstrated the requirement of
functional autophagy for lipid droplet biogenesis. In this work, authors have built on their
recently reported observations on the essential role of selective autophagy for lysosomal
degradation of NCoR1 transcriptional repressor. They have extended this work to show here
10 that a build-up of NCoR1 in autophagy deficient cells results in inhibition of LXR α transcription
factor with most striking consequences for lipid homeostasis. They have employed
complementary approaches in autophagy deficient models in vivo (hepatic mouse mutants) and
in cells to elegantly demonstrate that autophagic degradation of NCoR1 is essential for
expression of LXR α targets in fatty acid and TG synthesis as well as transport. The highlight of
15 this study is a demonstration that transient liver steatosis relies on autophagy in two
physiological metabolic and growth paradigms such as fasting and acute proliferation in
response to partial hepatectomy. Therefore, this study irrefutably demonstrated an essential
role of functional autophagy not only for lipid degradation but also for lipid synthesis and lipid
droplet biogenesis. In sum, it is an excellent work that exposes another functional facet of
20 autophagy in physiology. Most importantly, it also opens to future mechanistic questions on
how cells sense the needs in selective autophagy of transcriptional repressors to fine tune
activity and expression of transcription factors and the relevance of these mechanisms to
pathophysiology. The manuscript written with great style that makes it agreeable to follow the
story. Particularly, the introduction brings up known aspects of how autophagy affects lipid
25 metabolism and clearly frames the main question of this work. I have just few minor
suggestions.

Reply:

We would like to thank the Referee for the positive reception of our manuscript.

Minor comments:

30 Comment-1:

Add an explanation to the immunoblot quantifications (e.g. Fig.1A) about how exactly
comparisons are made (since relative values are presented should one condition be set as 1?).
Also, it is surprising that quantification of Fig.1A for cytosolic NCoR1 shows so little difference
35 unlike immunoblot capture that is presented.

Reply-1:

According to this comment, we revised the presentation and explained the immunoblot
quantification in detail (figure legends of Fig. 1A, Fig. 3A and Fig. 4A).

40 Fig. 1A: Signal intensities of cytoplasmic NCoR1 and Gapdh and nuclear NCoR1,
LXR α and Lamin B were measured by densitometry and were subtracted that of background.
Bar graphs indicate the average values of the indicated cytoplasmic and nuclear proteins relative
to Gapdh and Lamin B, respectively. The average value of the *Atg7^{fl/fl}* mice was set as 1.

45 Fig. 3A and 4A: Signal intensities of cytoplasmic NCoR1 and GAPDH and nuclear
NCoR1, LXR α and LAMIN B were measured by densitometry and were subtracted that of
background. Bar graphs indicate the average values of the indicated cytoplasmic and nuclear
proteins relative to GAPDH and LAMIN B, respectively. The average value of the
ATG7-knockout HepG2 expressing lacZ was set as 1.

Comment-2:

50 If text length allows, it would be interesting to discuss that although nuclear pool of NCoR1 is
protected from degradation in livers of autophagy deficient *Atg7LKO* mice, the fasting potently

induced degradation of cytosolic NCoR1 protein (judged from immunoblot presented on Fig.1A). Did total levels of NCoR1 change in livers of fasted mice?

55 Reply-2:

Thank you for the comment. As shown in Figure 1A of the revised manuscript, fasting down-regulated both nuclear and cytosolic NCoR1 levels in autophagy-deficient mouse livers. This may be attributed to the proteasomal degradation of NCoR1. Ubiquitination by a F-box-like/WD repeat-containing protein, TBLR1 directs NCoR1 into the proteasomal degradation and favors the exchange of corepressors for coactivators (PMID: 14980219, PMID: 18374649). This suggests that both the ubiquitin-proteasome and autophagy-lysosomal pathways contribute to degradation of NCoR1. We described such possibility in the result and discussion section of the revised manuscript (line 96-100 in the revised manuscript). We have not been unable to detect clear NCoR1 signal by immunoblot analysis with total homogenates due to high background.

Comment-3:

Describe in the text the differences in LXRA expression shown on Fig1A.

70 Reply-3:

Thank you for this comment. Actually, the level of LXRA in nuclear fraction of *Atg7*-deficient mouse livers was significantly lower than that of control livers (Fig. 1A). We described this result in the result and discussion section of the revised manuscript (line 111-113 in the revised manuscript).

75

Comment-4:

Harmonize the labelling of LXRA throughout the manuscript.

Reply-4:

80 In accordance with this suggestion, we unified the labeling. We described mouse gene in accordance with the International Committee on Standardized Genetic Nomenclature for Mice: Symbols begin with an uppercase letter followed by all lowercase letters except for recessive mutations, which begin with a lowercase letter.

85 Comment-5:

Revise figure legends e.g. Fig.1A (remove remark about total homogenates), Fig.S1 (change "hepatocytes" to "liver sections"). Specify what it means that experiments were performed 3 times (technical or biological replicates). Indicate in Material and method section timing conditions of fasting, sacrifice and surgical procedure in mice.

90

Reply-5:

Thank you for these comments. We corrected the figure legends including Fig. 1A and Fig. S1. We also described the detail experimental setting related to fasting, sacrifice and surgical procedure in the Material and methods section of the revised manuscript.

95

Comment-6:

100 *If text length allows, it would be important to bridge in discussion the findings on the role of autophagy in lipid synthesis and hepatocyte proliferation. To this end, an induction of FA synthesis is the most marked response to siNCoR1 in autophagy deficient HepG2 cancer cell line (Fig. 4B). From the other hand, hepatocyte proliferation and expression of enzymes of FA synthesis were decreased in *Atg7*LKO mice after hepatectomy (Fig.2). Notably, an increased de novo lipid synthesis is also a metabolic hallmark of proliferating cancer cells while decreased*

105 expression of NCoR1 was reported in liver cancer (PMID: 21075309). In lights of these findings, it is plausible that resistance of the Atg7LKO mice to progress from benign adenoma to liver cancer (PMID: 21498569) might be due to persistent expression of NCoR1 repressor. Therefore, these novel findings presented by Takahashi S. et al. might also be pertinent for liver pathophysiology.

Reply-6:

110 Thank you for this valuable suggestion. We discuss a possibility that the accumulation of NCoR1 in autophagy-deficient mouse livers prevents malignancy of the liver benign adenoma in the discussion section of the revised manuscript (line216-228 in the revised manuscript).

115 **Referee #2**

General comments:

120 *This Short Report by Takahashi et al reports on defective lipid droplet biogenesis and reduced hepatosteatosis in hepatocyte-specific Atg7 knockout mice. The authors have recently demonstrated that lipid oxidation was suppressed in the same mouse model (Saito et al., Nat Commun 2019). In this study, the suppression of lipid oxidation resulted from accumulation of nuclear receptor co-repressor 1 NCoR1, which suppressed PPARalpha activity and thereby decreased the expression of enzymes involved in beta-oxidation. In the current manuscript, the accumulation of NCoR1 is shown to result in the suppression of LXRalpha activity, decreasing the expression of enzymes involved in fatty acid and triglyceride synthesis. Overall, the current observations are in line with earlier findings and the known functions of LXRalpha in controlling lipogenic gene expression. NCoR1 is known to be a substrate for autophagy (Saito et al., 2019) and fine-tunes the transactivation of both PPARalpha and LXRalpha. Moreover, loss of autophagy initiation causes inactivation of both PPARalpha and LXRalpha (Ma et al., 2013). The present paper does not go beyond this mechanistically. The experiments are in general well executed and most of the results are sound. However, some of the major claims made are not well substantiated by the data. Please see specific points below.*

Reply:

135 We would like to thank the Referee for the crucial suggestions on how to improve our manuscript.

Comment-1:

140 *The NCoR1 silencing experiments (Fig. 4) do not actually support a major role for NCoR1 in lipid droplet formation. First, no differences in LDs are observed (Fig. 4C) and the effect on TAG levels is marginal, with some 10% increase. The authors suggest that this is because NCoR1 ablation enhances both the anabolism and catabolism of FAs. If this is the case, the authors might obtain clearer effects e.g. by manipulating FA catabolism in NCoR1 silenced cells. Second, the authors have strictly speaking not studied the formation or biogenesis of LDs. Rather, the steady-state numbers and sizes of LDs were analyzed. As LD maintenance is also dependent on LD breakdown, one cannot conclude that it is the formation of lipid droplets that is affected. Thus, the data do not convincingly show that NCoR1 is important for LD formation and do not support these statements in the title and abstract.*

Reply-1:

150 While tracer experiments with ¹³C-labeled Palmitate exhibited the inhibition of β-oxidation in autophagy-deficient mouse livers (Saito et al., *Nat Commun.*, 10, 1567, 2019), we have not provided the data directly showing that defective autophagy causes the impairment of lipogenesis and of LD formation, in this study. To this end, we conducted tracer experiments

155 with ¹³C-labeled glucose to evaluate the incorporation of glucose-derived ¹³C into Palmitic,
Oleic acid and Stearic acids (*i.e.*, evaluation of rate of lipogenesis) in autophagy-competent and
incompetent HepG2 cells. Though we detected ¹³C-incorporated fatty acids in both HepG2 cells,
we did not observe any significant differences among the cells (Figure 1 to the reviewers),
probably due to low detection sensitivity and/or incomplete experimental settings. Thus,
160 according to the editor's suggestion, we revised the title, abstract and text as adapting the
presented data.

Comment-2:

165 *The current findings are essentially opposite to those of Singh et al. (Nature 2009) that reports
on increased size and number of lipid droplets in hepatocytes lacking Atg7. The authors
speculate that the difference might be due to the different ages of animals: 5 weeks in the
present study as compared to four months in Singh et al. This should be possible to reconcile
easily, by aging the mice till 4 months and studying if the Atg7 deficient livers now become
steatotic.*

170 Reply-2:

In according to this suggestion, we investigated whether LDs accumulate in hepatocytes of
5-month old liver-specific *Atg7*-deficient mice. As shown in Supplementary Figure S1B in the
revised manuscript, we performed oil-red O staining with 5-month old *Atg7^{fl/fl}* and *Atg7^{fl/fl}*;
Alb-Cre mice and noticed that perivenous hepatocytes in 5 month-old *Atg7^{fl/fl}*; *Alb-Cre* mice
175 contained many LDs though totally lower level compared with of extent to age-matched *Atg7^{fl/fl}*
mice. We described the above-mentioned result in the result and discussion section of the
revised manuscript (line 213-215 in the revised manuscript).

Comment-3:

180 *The differences in lipid droplet numbers and sizes quantified from cells do not match well with
the images. For instance, the autophagy competent cells shown have many more lipid droplets
than ten (or less), as indicated in the bar graphs. The images look as if lipid droplets may have
fused (a common problem when using glycerol containing mounting media).*

185 Reply-3:

While we quantified the number and size of LDs using a CellInsight™ CX5 High-Content
Screening Platform (Thermo Fisher Scientific) using HCS Studio™ software, we obtained cell
images using a confocal laser-scanning microscope (Olympus, FV1000), which may have
caused an inconsistency between presented images and quantification. We replaced original
190 images with the representative ones.

Comment-4:

Figure 1 A repeats findings shown in Saito et al., 2019 and this should be clearly spelled out.

195 Reply-4:

We spelled out that Figure 1 is the confirmation of previous finding (line 92-94 in the revised
manuscript).

200 -----
Referee #3

General comments:

*Autophagy plays a role in both lipid droplet biogenesis and in lipid droplet turnover via
lipophagy. The mechanisms underlying the role of autophagy in LD biogenesis are not fully
understood. Recent work from Saito et al and Iershov et al (both 2019) have shown that*

205 autophagy can regulate lipid metabolism by promoting the turnover of NCoR1, a negative
regulator of PPAR α that induces expression of genes involved in fatty acid oxidation. However,
NCoR1 also regulates LXRA which promotes lipid synthesis, and thus acts counter to PPAR α .
To address this dichotomy, the authors examined NCoR1 levels and activity in the liver of
210 Alb-Cre:Atg7^{fl/fl} mice in which autophagy is impaired. Consistent with Saito and Iershov,
NCoR1 levels accumulated in the liver when ATG7 was knocked out and autophagy effectively
reduced.

Acly, Fasn, Scd1 and Acaca are all LXRA target genes and were down-regulated in Atg7
deleted livers. Fasting reduced their expression in WT but not in the KO. Other targets GK,
Dgat1, Fatbp2 that were upregulated in WT liver by fasting were not induced in the Atg7 KO
215 livers. Lxra mRNA expression was also down consistent with Lxra being autoregulated .
Basically, similar to effects on PPAR α , autophagy inhibition caused NCoR1 to accumulate and
to inhibit LXRA activity. Interestingly, lipid droplet numbers in fasted livers were reduced by
Atg7 deletion, based on ORO staining. Is this due to reduced formation of LDs? Suppression of
lipogenesis and Lxra target genes was also detected in response to partial hepatectomy
220 resulting in reduced liver size following regeneration. Knocking out ATG7 in HepG2 cells also
caused NCoR1 to accumulate and LXRA to be inhibited, as determined by levels of LXRA target
genes. This was rescued by knockdown of NCoR1 although this did not rescue the number and
size of lipid droplets since NCoR1 influences both FA oxidation (via PPAR α) and lipogenesis
(shown here via LXRA).

225 Overall, this work is highly complementary to the studies of Saito et al and Iershov et al and
extends them to show that autophagic regulation of NCoR1 also influences lipogenesis via
LXRA. It does leave the conundrum of how and when NCoR1 may differentially regulate PPAR α
and LXRA in a manner modulated by autophagy. An aspect of the work that could be enhanced,
is if the authors could measure rates of fatty acid oxidation and de novo lipogenesis in the
230 control and Atg7 deleted hepatocytes to confirm that autophagy is indeed impacting both these
processes via NCoR1 turnover. Work that is perhaps beyond the scope of this study is to
examine mutants of LXRA that cannot bind NCoR1 - are these mutants resistant to the effects of
autophagy inhibition. As it stands, the scope of the paper is limited but what is included is
useful and well controlled data.

235

Reply:

We would like to thank the Referee for the positive appreciation of the manuscript and
thoughtful advice: the measurement of rates of fatty acid oxidation and de novo lipogenesis, and
the experiments with LXRA mutant that is unable to bind to NCoR1.

240 Related to comment-1 together with comment-1 of Referee 2, while tracer
experiments with ¹³C-labeled Palmitate exhibited the inhibition of β -oxidation in
autophagy-deficient mouse livers (Saito et al., *Nat Commun.*, 10, 1567, 2019), we have not
provided the data directly showing that defective autophagy causes the impairment of
lipogenesis and of LD formation, in this study. To address this issue experimentally, we
245 conducted tracer experiments with ¹³C-labeled glucose and evaluated the incorporation of
glucose-derived ¹³C into Palmitic, Oleic acid and Stearic acids (i.e., evaluation of rate of
lipogenesis) in autophagy-competent and incompetent HepG2 cells. Though we detected
¹³C-incorporated fatty acids in both HepG2 cells, we did not observe any significant differences
among the cells (Figure 1 to the reviewers), probably due to low detection sensitivity and/or
250 incomplete experimental settings. Thus, according to the editor's suggestion, we revised the title
and abstract as adapting the presented data.

Regarding second comment, we agree that such experiment is useful, but we think
that it is beyond the scope of current study as this referee mentioned.

December 17, 2019

RE: Life Science Alliance Manuscript #LSA-2019-00513-TR

Dr. Masaaki Komatsu
Juntendo University School of Medicine
Department of Physiology
Hongo 2-1-1
Bunkyo-ku, Tokyo 113-8421
Japan

Dear Dr. Komatsu,

Thank you for submitting your revised manuscript entitled "Loss of autophagy impairs physiological steatosis by accumulation of NCoR1". As you will see, the reviewers appreciate the changes introduced in revision, and we would thus be happy to publish your paper in Life Science Alliance, pending final small revisions:

- Please address the remaining concern of reviewer #1 (see also last bullet point listed here)
- The name of co-author Dr. Hyeon-Cheol Lee-Okada in the submission system is not matching the one depicted in the manuscript (Hyeon-Cheol Lee), please clarify
- Please list 10 authors et al in the reference list
- Please provide less over-contrasted source data for the western blots in Fig 1A and S2
- For Figure legends 2-4, please correct the following sentence to improve clarity: "RT qPCR analyses were performed a technical replicate on each biological sample."

A. FINAL FILES:

B. MANUSCRIPT ORGANIZATION AND FORMATTING:

Sincerely,

Reviewer #2 (Comments to the Authors (Required)):

The concerns raised have, in principle, been satisfactorily addressed. Some sentences of the revised text (especially beginning of page 9, red text) should be double checked for clarity.

Reviewer #3 (Comments to the Authors (Required)):

The revised manuscript is significantly enhanced by the revisions carried out since I first reviewed it and I support publication of this work.

Editor's comments (italized)Comment-1:

Please address the remaining concern of reviewer #1 (see also last bullet point listed here).

5 Reply-1:

We corrected the sentence pointed out by the reviewer (Page 8, lines 208-210 of the revised manuscript).

10 Comment-2:

The name of co-author Dr. Hyeon-Cheol Lee-Okada in the submission system is not matching the one depicted in the manuscript (Hyeon-Cheol Lee), please clarify.

15 Reply-2:

"Hyeon-Cheol Lee" is correct. We replaced "Hyeon-Cheol Lee-Okada" by "Hyeon-Cheol Lee".

Commnet-3:

Please list 10 authors et al in the reference list.

20 Reply-3:

In accordance to style of Life Science Alliance, we corrected the reference list.

Commnet-4:

Please provide less over-contrasted source data for the western blots in Fig 1A and S2.

25 Reply-4:

We provided the source data.

Commnet-5:

30 *For Figure legends 2-4, please correct the following sentence to improve clarity: "RT qPCR analyses were performed a technical replicate on each biological sample."*

35 Reply-5:

Thank you for the comment. We corrected the figure legends including Fig. 2, 3 and 4 (Page 16, lines 460-462, page 17, lines 487-489 and page 18, lines 514-515 of the revised manuscript).

Referee's comments (italized)**Referee #2**Comment-1:

40 *The concerns raised have, in principle, been satisfactorily addressed. Some sentences of the revised text (especially beginning of page 9, red text) should be double checked for clarity.*

45 Reply-1:

Thank you for the comments. We corrected the sentence (Page 8, lines 208-210 of the revised manuscript).

Referee #3Comment-1:

50 *The revised manuscript is significantly enhanced by the revisions carried out since I first reviewed it and I support publication of this work.*

Reply-1:

We would like to thank the Referee for the positive appreciation of the manuscript.

December 18, 2019

RE: Life Science Alliance Manuscript #LSA-2019-00513-TRR

Dr. Masaaki Komatsu
Juntendo University School of Medicine
Department of Physiology
Hongo 2-1-1
Bunkyo-ku, Tokyo 113-8421
Japan

Dear Dr. Komatsu,

Thank you for submitting your Research Article entitled "Loss of autophagy impairs physiological steatosis by accumulation of NCoR1". It is a pleasure to let you know that your manuscript is now accepted for publication in Life Science Alliance. Congratulations on this interesting work.

*****IMPORTANT:** If you will be unreachable at any time, please provide us with the email address of an alternate author. Failure to respond to routine queries may lead to unavoidable delays in publication.*******

DISTRIBUTION OF MATERIALS:

Again, congratulations on a very nice paper. I hope you found the review process to be constructive and are pleased with how the manuscript was handled editorially. We look forward to future exciting submissions from your lab.

Sincerely,
